# Entropy-guided $k$-Guard Sampling for Long-Horizon Autoregressive Video Generation

## Abstract

Autoregressive (AR) architectures have achieved significant successes in LLM, inspiring explorations for video generation. In LLMs, top-$p$/top-$k$ sampling strategies work exceptionally well: language tokens have high semantic density and low redundancy, so a fixed size of token candidates already strike a balance between semantic accuracy and generation diversity. In contrast, video tokens have low semantic density and high spatio-temporal redundancy. This mismatch makes static top-k/top-p strategies ineffective for video decoders: they either introduce unnecessary randomness for low-uncertainty regions (static backgrounds) or stuck in early errors for high-uncertainty regions (foreground objects). Prediction errors will accumulate as more frames are generated and eventually severely degrade long-horizon quality. To address this, we propose Entropy-Guided $k$-Guard (ENkG) sampling, a simple yet effective strategy that adapts sampling to token-wise dispersion, quantified by the entropy of each token's predicted distribution. ENkG uses adaptive token candidate sizes: for low-entropy regions, it employs fewer candidates to suppress redundant noise and preserve structural integrity; for high-entropy regions, it uses more candidates to mitigate error compounding. ENkG is model-agnostic, training-free, and adds negligible overhead. Experiments demonstrate consistent improvements in perceptual quality and structural stability compared to static top-k/top-p strategies.

## 1 Introduction

The field of video-based world models has witnessed explosive growth in recent years, with significant advancements in generating high-fidelity, temporally coherent, and physically plausible video sequences Villegas et al. (2022); Wang et al. (2024a). These models aim to build an internal representation of the world's dynamics, enabling applications from realistic simulation for robotics to advanced content creation He et al. (2025). This progress has paved the way for models that can not only synthesize video from text but also begin to understand and simulate interactive environments Mo et al. (2025).

Among the various architectural paradigms, autoregressive (AR) models have become a cornerstone for video generation . By factorizing the joint probability distribution of video frames into a product of conditional probabilities, AR models excel at capturing temporal causality and allow for fine-grained, frame-by-frame control during generation . This sequential approach is inherently flexible, supporting variable-length video generation and compatibility with scalable transformer architectures (Dosovitskiy et al., 2020; Weissenborn et al.). However, the sequential nature of AR models also introduces significant challenges: error accumulation (Parthipan et al., 2024; Bengio et al., 2015) and explosure bias (Schmidt, 2019). Minor inaccuracies or suboptimal choices in generating a single frame can propagate and amplify over time, leading to a degradation of quality, loss of coherence, and "drifting" from the intended content in longer video sequences (Hu et al., 2023).

Several strategies have been proposed to mitigate this effect. Huang et al. (2025) simulate inference during training by feeding the model its own previous predictions, allowing it to learn to correct mistakes. Other works introduce noisy or masked contexts, encouraging the model to be robust to imperfect inputs (Ren et al., 2025a; Zhou et al., 2025). While effective, these approaches often require modifications to the model architecture or additional training complexity, which may limit their applicability to existing large-scale video generation models.

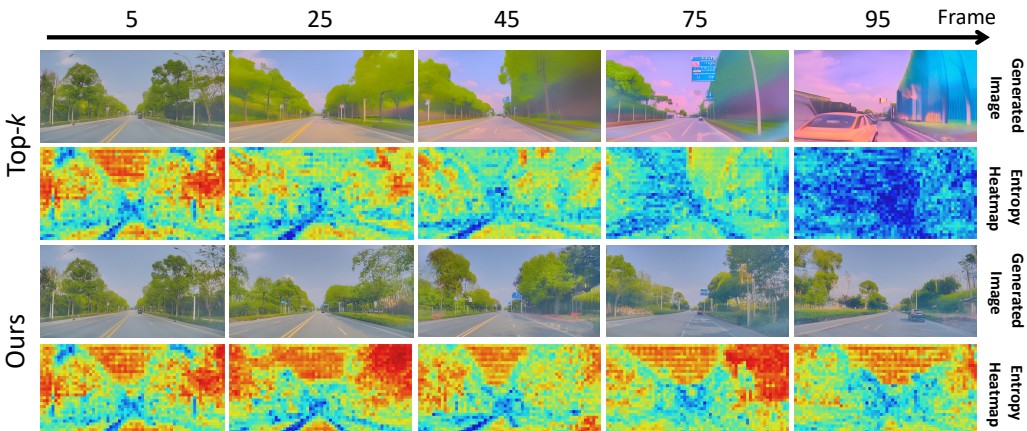

Figure 1: The results illustrate the phenomenon of entropy collapse in standard AR decoding, where blue regions indicate low entropy and red regions indicate high entropy. Our method effectively alleviates this issue.

In contrast, we focus on the often-overlooked role of the ***sampling process*** in autoregressive video generation. Our analysis reveals that conventional strategies such as fixed top-$k$ or nucleus (top-$p$) sampling fail to account for the spatially structured uncertainty inherent in video tokens. Specifically, we observe that high-entropy regions, corresponding to complex textures like foliage or road markings, are prone to brittleness, whereas low-entropy regions representing structured geometry can suffer from overconfidence and texture wash-out. This motivates an ***adaptive*** sampling policy that modulates token diversity based on entropy, effectively balancing stability and richness in generated content.

Specifically, we introduce **Entropy-guided $k$-Guard sampling**, a model-agnostic algorithm that dynamically adjusts the size of candidates for each token according to its entropy. First, our method measure the entropy of each video token that indicates the dispersion of video token probability. For low-entropy regions, our strategy employs fewer candidates to suppress redundant noise and preserve structural integrity; for high-entropy regions, it uses more candidates to mitigate error compounding. Unlike previous solutions that modify training or rely on multiple candidate evaluations, our approach operates purely at the inference stage, making it widely applicable to existing autoregressive video models.

We validate our method on several state-of-the-art autoregressive video generation architectures, demonstrating that it significantly reduces error accumulation, preserves fine-grained textures, and improves temporal coherence over extended sequences. Quantitative metrics and qualitative results confirm that our adaptive sampling strategy enables longer, more realistic video generation without retraining or architectural changes. These findings suggest that carefully designed inference-time strategies can be a powerful tool for improving autoregressive video generation, complementing existing advances in model design and training.

In summary, our contributions are threefold: (i) we identify the limitations of fixed sampling strategies in autoregressive video generation and highlight the role of spatially structured uncertainty in error accumulation, (ii) we propose a simple yet effective entropy-guided adaptive sampling strategy with a $k$-guard mechanism, and (iii) we empirically demonstrate that this method improves long-sequence video quality across multiple benchmark models. This work highlights the potential of uncertainty-aware inference as a practical and generalizable solution for high-fidelity video synthesis.

## 2 RELATED WORK

### 2.1 VIDEO WORLD MODELS

Video-based world models aim to learn an internal representation of an environment, allowing the system to predict future states, simulate interactions, and support planning (Ha & Schmidhuber,

2018; Ding et al., 2024; Long et al., 2025; Zhang et al., 2025). Recent progress in large-scale video generation has enabled the creation of high-fidelity world simulators capable of producing visually realistic and physically plausible sequences (OpenAI, 2024), which are particularly valuable for applications such as autonomous driving and robotics (Wang et al., 2024b; Li et al., 2025).

A wide range of generative architectures have been explored for video synthesis. Diffusion models and Generative Adversarial Networks (GANs) have shown success in producing high-quality frame sequences (Ho & Jain, 2022; Clark & Fidler, 2019), while autoregressive (AR) models have emerged as a powerful alternative due to their inherent capacity to model temporal dependencies. By factorizing the joint distribution of video tokens or frames into a product of conditional probabilities, AR models generate coherent, long-duration videos either frame-by-frame (Gu et al., 2025) or token-by-token (Wu et al., 2024), enabling fine-grained temporal control and interactive generation.

Despite their strengths, AR models suffer from a fundamental limitation: *error accumulation*. During inference, each predicted frame or token is fed back as input for subsequent steps, so any imperfection can propagate and amplify over time (Yu & Chen, 2024; Feng & Li, 2021; Walker & Gupta, 2016). This leads to quality degradation, manifesting as flickering, unnatural motion, or drift from the intended scene (Yu & Chen, 2024; Saxena & Kumar, 2024; Benjamin & Smith, 2018; Kong et al., 2025). Error accumulation has been identified as a core challenge in theoretical analyses of autoregressive video generation, alongside issues such as memory bottlenecks (Saxena & Kumar, 2024; Goyal & Lee, 2022).

To address this limitation, our work proposes a *token-level adaptive sampling* strategy that dynamically modulates the sampling distribution according to the model's predictive uncertainty. By integrating uncertainty into the generation process, we directly mitigate the compounding of errors, preserving both temporal coherence and visual fidelity in long-duration video sequences.

## 2.2 Autoregressive Sampling Algorithms

Sampling strategies are central to autoregressive (AR) generation. *Greedy decoding* selects the most likely token at each step, but often yields low diversity. *Beam search* explores multiple hypotheses in parallel and improves likelihood-based metrics, yet typically reduces diversity. A widely used family of stochastic methods is *truncated probability sampling*, including top-$k$ (Noarov et al., 2025) and nucleus (top-$p$) sampling Ravfogel et al. (2023). Both restrict sampling to a subset of the distribution, balancing diversity and quality, but their hard truncation can occasionally introduce rare erroneous tokens, causing catastrophic errors in long sequences or generate duplicate and over-confident content with low threshold. *Best-of-$N$ sampling* (Snell et al., 2024) generates multiple candidates and selects the best according to a reward model or predefined metric. While effective, it operates at a coarser granularity, is model-dependent, and incurs significant additional computation. Recently, entropy-based strategies have been explored in large language models (LLMs), where entropy guides model size switch (Simonds, 2025) or retrieval augmentation (Qiu et al., 2025). Similar entropy-guided temperature scaling methods have been proposed for LLMs (Zhang et al., 2024) and image generation (Ma et al., 2025) to balance diversity and fidelity. In contrast to these temperature-scaling approaches, our method leverages entropy to adaptively adjust the Top-$p$ threshold with a $k$-guard mechanism. Crucially, we address the video-specific challenge of temporal error accumulation, rather than the static trade-offs focused on in text and image modalities.

## 3 Motivation

We identify three critical findings in autoregressive (AR) video generation models. These observations explain why static sampling (effective for LLMs) fails for video and therefore motivate our dispersion-aware adaptive sampling strategy.

**F1. Video token probability distributions are inherently flat.** As shown in Figure **??**, generated video tokens usually have quite small probability values, where dozens of tokens could achieve total probability. Tiny gaps between top candidates mean small logit perturbations (model noise, temporal drift) easily flip the argmax, breaking temporal coherence and accumulating visual artifacts. Truncated sampling eases brittleness but remains one-size-fits-all.

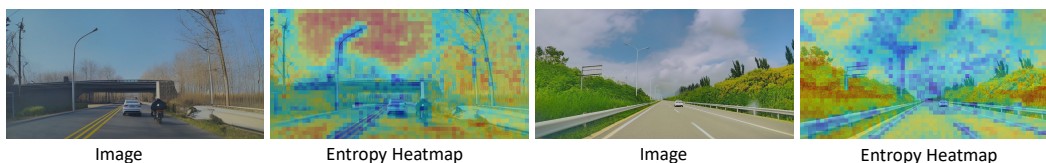

| Image | Entropy Heatmap | Image | Entropy Heatmap |

Figure 2: The visualization of entropy heatmaps. High-entropy regions form repeating textures (e.g., sky, foliage, and road), while low-entropy regions cluster in structured content and distinguishable textures (e.g., boundaries, edges between sky and trees, road markers and lines).

The root cause lies in fundamental differences between video and language tokens. Language tokens, with high semantic specificity (each mapping to a clear meaning, e.g., "car") and low redundancy (rarely repeating adjacent tokens). The language tokens have sharp probability distributions where top-1 probabilities often reach 0.7–0.8 (e.g., "street" in "I walk along the ___"). In contrast, video tokens lack direct semantic grounding and have high spatiotemporal redundancy. With no single token carrying unique meaning, their distributions are flat and diffuse, with an average top-1 probability of just 0.2.

**F2. Video token probability dispersion is inherently tied to the semantic structure of the image**. Given an AR model, the predicted distribution at the token $i$ is $P_i$, and the entropy is as follows.

$$H_i = -\sum_j p_i(j) log p_i(j) \tag{1}$$

where $j$ is the vocabulary index.

Entropy is a tool to measure the model's uncertainty Stolfo et al. (2024); Kang et al. (2025) and the token probability dispersions in predictions. Higher entropy indicates low confidence, thus showing greater dispersion in distributions, while lower entropy (high confidence) presents more concentrated distributions. Figure 1 demonstrates how the entropy heatmaps links the generation quality. High-dispersion (high-entropy) regions form repeating textures (e.g., sky, foliage, and road), where multiple tokens are equally plausible due to subtle texture variations. In contrast, low-dispersion (low-entropy) regions cluster in structured content and distinguishable textures (e.g., boundaries, edges between sky and trees, road markers and lines), where only a few tokens match the stable pixel patterns. However, existing static sampling (top-k or top-p) ignores this structure, forcing large candidate pools on low-dispersion regions (introducing redundant noise) and small pools on high-dispersion ones (discarding valid tokens), thereby exacerbating error accumulation.

**F3. *Entropy Collapse* in Long-Horizon Autoregressive Video Generation.** A third critical finding is that AR video models suffer from entropy collapse during long-horizon generation—an issue tied to evolving token dispersion patterns. As shown in Figure 1, this collapse manifests in two ways: temporally, the share of low-dispersion (low-entropy) tokens grows rapidly with each frame, driving down frame-averaged entropy; spatially, low-dispersion regions expand outward, gradually encroaching on high-dispersion areas, which erodes fine textures (e.g., foliage, road cracks) and replaces them with oversmoothed, uniform blocks (e.g., a detailed tree reduced to solid green). This collapse stems from the model overcommitting to low-dispersion token choices as generation proceeds, which reinforces structural drift and texture wash-out. Notably, entropy collapse is unique to video generation: LLMs avoid it because the high semantic density of language tokens prevents overconfidence in redundant sequences.

**Insights.** To address these challenges, we propose a locally adaptive, entropy-guided sampling strategy: align candidate pool size with token dispersion. For low-dispersion regions, small pools (with a minimal guard-n) suppress redundant noise and prevent entropy collapse, preserving structural stability while retaining baseline stochasticity. For high-dispersion regions, large pools include all plausible tokens to avoid brittle argmax flips and mitigate early error accumulation. The minimal guard-n is key—it avoids the extremes of greedy decoding (accelerating texture wash-out) and over-large pools (introducing noise), balancing stability and richness. Details of the entropy-to-k mapping for efficient implementation are provided in Section 4.

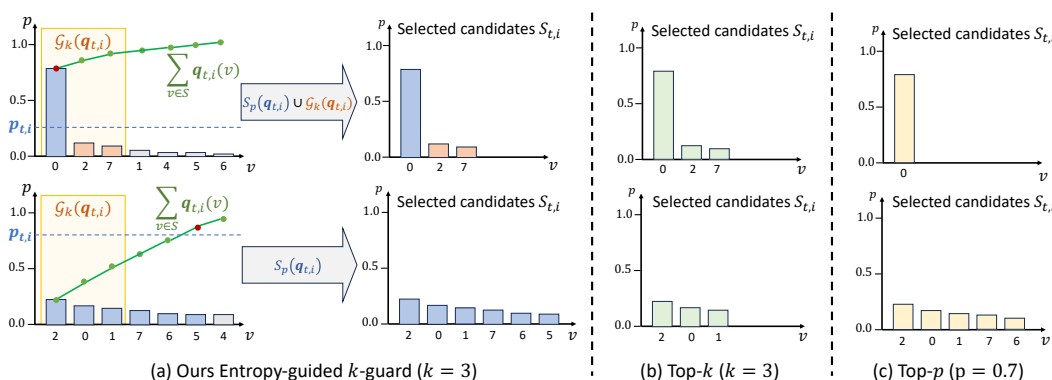

Figure 3: The overall illustration of our sampling strategy.

## 4 METHOD

Inspired by the findings in Section 1, we propose an Uncertainty-aware Adaptive Sampling strategy. The core idea is to leverage the model's predictive uncertainty at each token to dynamically guide the sampling process. Specifically, for regions where the model is confident, we enforce a conservative, near-greedy sampling to preserve structure. Conversely, for ambiguous regions, we encourage more diversity to mitigate brittle decisions and enrich textures. This is implemented through a three-stage process: (1) quantifying token-wise uncertainty using entropy, (2) mapping this uncertainty to an adaptive nucleus threshold, and (3) incorporating a "k-guard" to ensure robust exploration. We provide a pseudocode in Alg. 1.

### 4.1 PRELIMINARY

We begin by formalizing the **autoregressive (AR) formulation** for video generation. Let $\mathcal{V}$ denote a discrete codebook of size $V$, obtained via a learned tokenizer such as VQ-VAE. Each video frame is represented as a grid of tokens $\{z_{t,i} \in \mathcal{V}\}$, where $t$ indexes the temporal step and $i \in \{1, 2, \ldots, m\}$ indexes the spatial positions within a frame (each frame contains $m$ VQ tokens).

An AR world model factorizes the joint distribution of tokens as a product of conditional probabilities. Specifically, the probability of generating the $i$-th token in frame $t$ is conditioned on all previously decoded tokens in the same frame, as well as historical context and actions:

$$p(z_{t,i} \mid z_{t,<i}, c_{<t}, a_{<t}) = \prod_{j=1}^{i} p(z_{t,j} \mid z_{t,j-1}, c_{<t}, a_{<t}), \tag{2}$$

where $z_{t,<i}$ denotes the previously decoded tokens within the current frame, $c_{<t}$ represents observed context such as conditioning frames, and $a_{<t}$ denotes historical actions. When $i = m$, the above product gives the joint probability of generating the entire $t$-th frame.

### 4.2 INSTABILITY IN AUTOREGRESSIVE VIDEO GENERATION

However, autoregressive token generation is inherently prone to instability due to **error accumulation**. Let $\hat{z}_{t,j}$ denote the token actually generated at step $j$. Then the conditional probability for the next token depends on previously generated (potentially erroneous) tokens:

$$p(z_{t,i} \mid \hat{z}_{t,<i}, c_{<t}, a_{<t}) \neq p(z_{t,i} \mid z_{t,<i}, c_{<t}, a_{<t}), \tag{3}$$

where $\hat{z}_{t,<i}$ contains tokens that may differ from the ground truth $z_{t,<i}$. Consequently, small errors propagate through the sequence, amplifying discrepancies in later tokens and potentially degrading entire frames.

During inference, the model generates a video sequence by sequentially sampling tokens from these categorical distributions. The choice of sampling strategy is therefore critical: greedy decoding

Figure 4: Visual results of DrivingWorld and VaVIM models with our strategy..

---

**Algorithm 1** ENkG Sampling

---

**Require:** probability distribution $\mathbf{p} \in \mathbb{R}^V$; hyperparameters $(\alpha, \beta, p_{\text{low}}, p_{\text{high}}, k_{\text{g}})$
**Ensure:** Sampled token indices $\mathbf{y} \in \mathbb{Z}$
1: Compute normalized entropy: $\mathcal{H} \leftarrow -\frac{1}{\log V} \sum_i p_i \log p_i$
2: Map entropy to nucleus probability via affine and clip:

$$p_{t,i} \leftarrow \text{clip}\big(\alpha\, \widehat{\mathcal{H}}_{t,i} + \beta,\ p_{\text{low}},\ p_{\text{high}}\big)$$

3: Let $\{q_{(i)}\}_{i=1}^V$ be probabilities sorted in descending order
4: Find cutoff $c = \min\{j : \sum_{i=1}^j q_{(i)} \geq p\}$
5: Set $c \leftarrow \max(c, k_{\text{g}})$
6: Define truncated distribution $\tilde{q}_i = \frac{q_{(i)}}{\sum_{j=1}^c q_{(j)}}$ for $i \leq c$, $0$ otherwise
7: Sample token $y \sim \tilde{q}$
8: **return y**

---

often produces blurry frames or repetitive collapse, while excessively random sampling amplifies noise and disrupts temporal coherence. To address this limitation, we introduce an **Entropy-Guided k-Guard (ENkG) sampling strategy** that dynamically adjusts sampling diversity according to the model's predictive confidence, balancing structural fidelity and the richness of textures.

### 4.3 ENTROPY-GUIDED K-GUARD SAMPLING

To quantify token-level uncertainty, we consider the predicted categorical distribution $q_{t,i}$ for each token $z_{t,i}$ at image token site $(t, i)$:

$$q_{t,i}(v) := p\big(z_{t,i} = v \mid z_{<t},\, z_{t,<i},\, c_{\leq t},\, a_{<t}\big), \quad v \in \mathcal{V}, \tag{4}$$

where $\mathcal{V}$ denotes the discrete codebook. The uncertainty associated with this prediction is measured by its Shannon entropy:

$$\mathcal{H}_{t,i} = -\sum_{v \in \mathcal{V}} q_{t,i}(v) \log q_{t,i}(v). \tag{5}$$

To obtain a standardized measure on the unit interval, we normalize the entropy by the maximum possible value $\log |\mathcal{V}|$:

$$\widehat{\mathcal{H}}_{t,i} = \frac{\mathcal{H}_{t,i}}{\log |\mathcal{V}|} \in [0, 1]. \tag{6}$$

Table 1: Quantitative results on Saturn and Nuplan. * *Cosmos uses a fixed 33-frame generation window; hence its metrics are computed on the first 33 frames (vs. 75 for others).*

| Model | DiverseDrive | | | | | Nuplan | | | | |
|---|---|---|---|---|---|---|---|---|---|---|
| | $FVD_{75}\downarrow$ | $FID_{75}\downarrow$ | LPIPS$\downarrow$ | PSNR$\uparrow$ | SSIM $\uparrow$ | $FVD_{75}\downarrow$ | $FID_{75}\downarrow$ | LPIPS$\downarrow$ | PSNR$\uparrow$ | SSIM $\uparrow$ |
| DrivingWorld(top-$k$ 30) | 696 | 61.78 | 0.401 | 14.03 | 0.43 | 583 | 37.80 | 0.380 | 14.22 | 0.39 |
| DrivingWorld(+Ours) | **489** | **26.61** | **0.350** | **15.87** | **0.45** | **565** | **31.34** | **0.360** | **14.96** | **0.40** |
| VaVIM(greedy) | 1473 | 91.75 | 0.396 | 16.46 | 0.50 | 927 | 65.26 | 0.315 | 14.82 | 0.44 |
| VaVIM(+Ours) | **1055** | **46.76** | 0.426 | 14.76 | 0.46 | 1031 | **41.60** | 0.327 | 14.43 | 0.42 |
| Cosmos(top-$p$ 0.8)* | 1260 | 87.82 | 0.48 | 16.56 | 0.54 | 814 | 80.45 | 0.29 | 17.52 | 0.54 |
| Cosmos(+Ours)* | **1132** | **84.67** | **0.47** | **16.61** | 0.53 | **801** | **75.01** | 0.29 | **17.81** | **0.55** |

The normalized entropy $\widehat{\mathcal{H}}_{t,i}$ serves as a direct indicator of the model's confidence in predicting token $z_{t,i}$. Low values of $\widehat{\mathcal{H}}_{t,i}$ correspond to sharply peaked distributions, indicating high confidence in a dominant token, whereas high values indicate flatter distributions and thus greater uncertainty. Specifically, $\widehat{\mathcal{H}}_{t,i} \approx 0$ corresponds to a highly confident, nearly deterministic prediction, while $\widehat{\mathcal{H}}_{t,i} \approx 1$ corresponds to a nearly uniform, highly uncertain prediction.

**Entropy-guided adaptive nucleus.** To adaptively control sampling diversity, the normalized predictive entropy $\widehat{\mathcal{H}}_{t,i}$ is mapped to a target cumulative probability $p_{t,i} \in [p_{\text{low}}, p_{\text{high}}]$ via an affine transformation with clipping:

$$p_{t,i} = \text{clip}\Big(\alpha\,\widehat{\mathcal{H}}_{t,i} + \beta,\ p_{\text{low}},\ p_{\text{high}}\Big).$$
$$\textbf{where } \alpha = \frac{p_{\text{high}} - p_{\text{low}}}{\widehat{\mathcal{H}}_{\text{high}} - \widehat{\mathcal{H}}_{\text{low}}}, \beta = p_{\text{low}} - \alpha\,\widehat{\mathcal{H}}_{\text{low}}. \tag{7}$$

where $\text{clip}(x, a, b) = \min(\max(x, a), b)$. In the experiments, $p_{\text{low}} = 0.65$, $p_{\text{high}} = 0.9$, and $\widehat{\mathcal{H}}_{\text{low}} = 0.25$, $\widehat{\mathcal{H}}_{\text{high}} = 0.6$.

Based on $p_{t,i}$, the adaptive nucleus set $\mathcal{S}_p(\boldsymbol{q}_{t,i})$ is defined as the minimal subset of tokens whose cumulative probability meets or exceeds $p_{t,i}$:

$$\mathcal{S}_p(\boldsymbol{q}_{t,i}) = \arg\min_{\mathcal{S} \subseteq \mathcal{V}} \Big\{ |\mathcal{S}| \ \Big| \ \sum_{v \in \mathcal{S}} q_{t,i}(v) \geq p_{t,i} \Big\}. \tag{8}$$

Tokens with low entropy correspond to high-confidence predictions, allowing a small nucleus and near-greedy sampling that preserves fine structures such as edges and boundaries. High-entropy tokens indicate greater uncertainty; a larger nucleus in these cases encourages exploration, enhances diversity, and mitigates the compounding of errors in sequential autoregressive generation. This entropy-guided adaptation provides a principled mechanism to balance fidelity and diversity in token-level sampling.

$k$**-Guard for robust exploration.** Direct entropy-guided adaptive sampling can become overly greedy in high-confidence (low-entropy) regions, where the nucleus set may contain only a few tokens. To preserve minimal exploration without compromising stability, the nucleus is augmented with the $k_g$ most probable tokens, forming a $k$-guard:

$$\mathcal{S}_{t,i} = \mathcal{S}_p(\boldsymbol{q}_{t,i}) \ \cup \ \mathcal{G}_k(\boldsymbol{q}_{t,i}, k_g), \tag{9}$$

where $\mathcal{S}_p(\boldsymbol{q}_{t,i})$ denotes the nucleus (top-$p$) candidate set, and $\mathcal{G}_k(\boldsymbol{q}_{t,i}, k_g)$ returns the indices of the $k_g$ tokens with the highest probabilities under $\boldsymbol{q}_{t,i}$. Typical choices for $k_g$ are small integers such as 3, 5, or 10. If the nucleus already contains these tokens, the union leaves the set unchanged. To limit computational cost, a maximum size $n_{\max}$ can be enforced by retaining only the top $n_{\max}$ tokens sorted by probability.

Token selection is then performed by sampling from the renormalized distribution over $\mathcal{S}_{t,i}$. This combined framework leverages token-wise uncertainty to adaptively regulate sampling diversity, while the $k$-guard ensures minimal exploration in highly confident regions, enhancing the robustness and stability of sequential autoregressive generation.

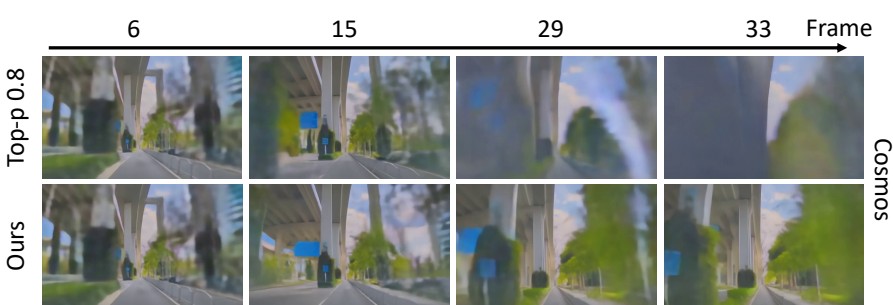

Figure 5: Visual results of Cosmos model with our strategy.

Table 2: Ablation study with DrivingWorld model on self-collected dataset.

| Method | FVD↓ | FID↓ | LPIPS↓ | PSNR↑ |
|---|---|---|---|---|
| Full Strategy | **489** | **26.61** | **0.350** | **15.87** |
| w/o Entropy | 532 | 41.43 | 0.591 | 13.96 |
| w/o k-Guard | 552 | 39.76 | 0.421 | 15.18 |

## 5 EXPERIMENTS

### 5.1 EXPERIMENTAL SETTING

**Models.** Since ENkG is a plug-and-play solution, we integrate it with existing AR-based video world model for experiments, including DrivingWorld (Hu et al., 2024), VaVIM (Bartoccioni et al., 2025) and Cosmos (NVIDIA, 2025). We keep generation parameters consistent with the original model for fair comarision. Specifically, DrivingWorld adopts top-$k(k = 30)$, VaVIM adopts greedy sampling, while Cosmos uses a top-$p$ ($p = 0.8$).

**Evaluation Dataset.** We conduct evaluations on two datasets: DiverseDrive and nuPlan. DiverseDrive is a self-collected high-quality driving dataset, which consists of 50 video clips. Compared to nuPlan datasets, DiverseDrive contains more scenarios and a richer variety of plants. These characteristics promote stronger generalization, making DiverseDrive a closer match to the requirements of world-model evaluation.

**Metrics.** To assess the quality of generated videos, we report the Fréchet Video Distance (FVD) as a measure of video-level realism, and the Fréchet Inception Distance (FID) to evaluate per-frame image fidelity. In addition, we include low-level metrics such as LPIPS, PSNR, and SIIM as supplementary evaluations, though these metrics are not well-suited for the video generation task.

### 5.2 MAIN RESULTS

**Quantitative Comparison.** As shown in Table 1, integrating ENkG consistently yields substantial gains across different architectures. On DiverseDrive, our method reduces FVD and FID by an average of 22.8% and 36.5% respectively, while also lowering LPIPS and improving PSNR/SSIM, indicating both perceptual and structural benefits. DrivingWorld also benefits from ENkG on Nuplan, despite being sufficiently trained on this dataset. Even Cosmos, which has a relatively weak AR backbone, achieves modest improvements on DiverseDrive. Notably, VaVIM tends to generate repeated frames, which artificially yields relatively lower FVD values; our strategy, by contrast, effectively alleviates this frame-freezing issue.

**Qualitative Comparison.** As shown in Fig. 4 and Fig. 5, the existing sampling techniques frequently result in textural degradation, where crucial details in road markings, such as crosswalks, and surrounding vegetation become indistinct and blurry. Furthermore, these approaches are susceptible to color distortion, leading to unnatural and washed-out hues that compromise the scene's realism. The generated sequence of VaVIM collapses into a static or near-static frame, failing to capture the inherent dynamics of the driving environment and rendering vehicles motionless. In contrast, our entropy-guided approach, which dynamically adjusts the size of candidates to prevent overconfidence, demonstrates substantial improvements in perceptual quality. Our strategy effectively mitigates the aforementioned issues, producing videos with sharp, well-defined textures and accurate color fidelity.

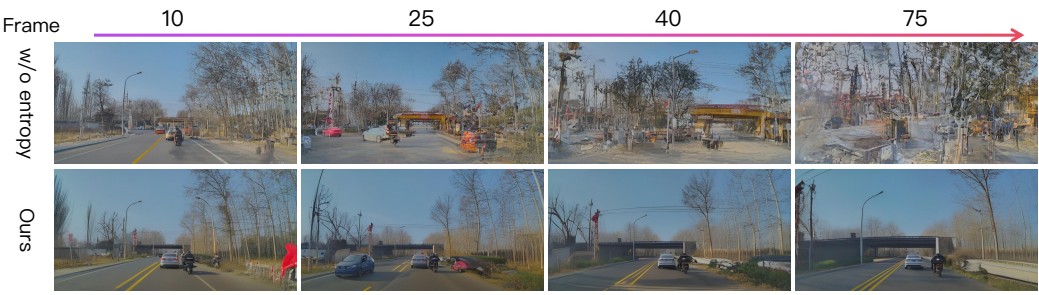

Figure 6: Entropy-adaptive guidance prevents collapse in the video model.

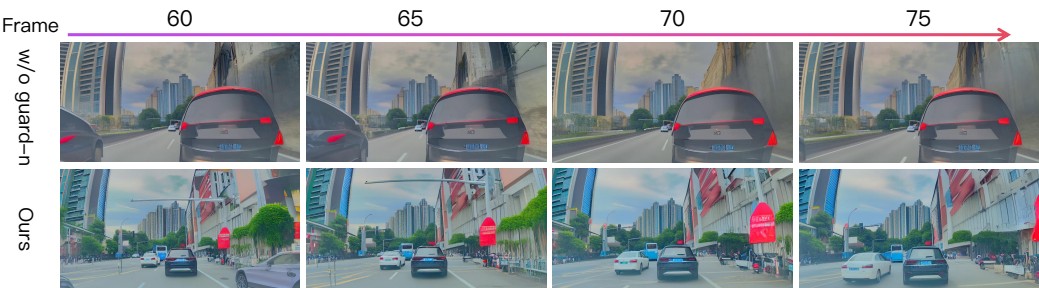

Figure 7: The k-guard design prevents frame-freezing in the video model.

## 5.3 ABLATION STUDY

**Effect of entropy-adaptive guidance.** The core contribution of the entropy-adaptive guidance is the dynamic adjustment of the sampling nucleus based on the model's predictive uncertainty. As shown in Figure 1 and Figure 6, it effectively mitigates issues of textural decay and color shifting commonly seen in baseline methods. By allowing a wider range of tokens when uncertainty is high and narrowing it when the model is confident, our method preserves high-frequency details, resulting in significantly sharper textures on surfaces like road markings and vegetation. This enhancement in per-frame visual fidelity translates to lower FID and LPIPS scores, indicative of more realistic and perceptually similar generated frames. Consequently, entropy-adaptive guidance significantly improves the visual quality and realism of the generated videos.

**Effect of $k$-guard.** The $k$-guard mechanism ensures a minimum level of diversity in candidate tokens. Without the $k$-guard, the model can, even with entropy-adaptive guidance, become overly confident in certain contexts. In Figure 7, this leads to significant temporal artifacts, such as vehicles that remain nearly stationary when they should be in motion, a physically implausible scenario. The introduction of $k$-guard directly addresses this failure mode, leading to more fluid and realistic motion dynamics, as quantitatively reflected in the substantial reduction of the FVD score, which measures temporal consistency. Therefore, the $k$-guard is crucial for maintaining temporal coherence and preventing the generation of degenerate, static sequences.

## 6 CONCLUSION

In this work, we investigated the challenge of error accumulation in autoregressive video generation and highlighted the overlooked role of the sampling process. We proposed *Uncertainty-aware Adaptive Sampling*, a simple yet effective strategy that dynamically modulates token diversity based on predictive entropy with a minimal $k$-guard. Unlike prior approaches that require architectural changes or retraining, our method operates purely at inference time, making it broadly applicable to existing large-scale models. Extensive experiments demonstrate that our approach significantly improves temporal coherence, preserves fine-grained details, and extends the effective generation horizon. These results suggest that inference-time uncertainty-aware strategies provide a practical and generalizable path toward more robust and high-fidelity video world models.

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

# A  APPENDIX

## A.1  EXPERIMENTAL PROTOCOLS (CONCISE)

We report FVD, FID, LPIPS, PSNR, and SSIM under identical preprocessing and frame sampling across methods unless noted. **Cosmos note:** Cosmos uses a fixed 33-frame generation window; thus all Cosmos metrics are computed on the first 33 frames of each sequence, while others use 75 ($\text{FVD}_{75}$, $\text{FID}_{75}$).

## A.2  USE OF LLMs (DISCLOSURE)

LLMs were used for language polishing and minor code refactoring suggestions only; authors verified all outputs. No dataset labeling or claim-critical content was delegated without human validation.

## A.3  ETHICS (BRIEF)

We comply with dataset licenses and anonymization policies; no personally identifiable information was used. Potential risks (e.g., misuse in safety-critical scenarios) are discussed and bounded by research-only usage.

# B  HYPERPARAMETER SENSITIVITY AND BASELINE TUNING

This section provides additional analyses on the sensitivity of ENkG to its hyperparameters and on the tuning of static sampling baselines. Unless otherwise noted, all experiments are conducted on DrivingWorld.

## B.1  SENSITIVITY OF ENkG HYPERPARAMETERS

**Robustness to entropy and probability thresholds.**  We first study the robustness of ENkG with respect to the entropy thresholds ($H_{\text{low}}, H_{\text{high}}$) and the corresponding probability range ($p_{\text{low}}, p_{\text{high}}$) used for entropy-guided truncation. Starting from the default configuration (*Mid*), we construct two extreme variants: a conservative configuration (*Left*), which prefers smaller $p$ and narrower entropy range, and an aggressive configuration (*Right*), which allows larger $p$ and a wider entropy range.

Table 3 shows that all metrics remain stable across these settings. Although the extreme variants cause moderate degradation in FVD/FID, the overall differences are small, indicating a broad performance plateau. This suggests that ENkG does not require delicate hand-tuning of entropy or probability thresholds to remain effective.

Table 3: Sensitivity to entropy thresholds ($H_{\text{low}}, H_{\text{high}}$) and probability band ($p_{\text{low}}, p_{\text{high}}$). The default configuration (*Mid*) achieves the best trade-off, while both conservative (*Left*) and aggressive (*Right*) shifts only moderately affect the performance.

| Setting | $H_{\text{low/high}}$ | $p_{\text{low/high}}$ | FVD $\downarrow$ | FID $\downarrow$ | LPIPS $\downarrow$ | SSIM $\uparrow$ | PSNR $\uparrow$ |
|---|---|---|---|---|---|---|---|
| Left | 0.0 / 0.5 | 0.60 / 0.90 | 522.05 | 30.93 | 0.35 | 0.49 | **16.26** |
| **Mid (Default)** | **0.25 / 0.60** | **0.65 / 0.90** | **489.00** | **26.61** | **0.35** | **0.45** | 15.87 |
| Right | 0.40 / 0.90 | 0.80 / 0.95 | 497.52 | 29.91 | 0.35 | 0.48 | 15.98 |

**Insensitivity to guard size $k_g$.**  Next, we fix all other ENkG hyperparameters and vary the guard size $k_g$, which controls how many top candidates are preserved by the $k$-guard at each step. As shown in Table 4, the performance is highly stable for any $k_g \in [2, 15]$, and only the degenerate case $k_g = 1$ (which effectively disables the guard) leads to a clear degradation. This indicates that while the *presence* of the $k$-guard is crucial to prevent collapse, its exact value is not sensitive in a wide range.

Table 4: Sensitivity to guard size $k_g$ (other hyperparameters fixed). ENkG remains stable for a broad range of $k_g$, and only the degenerate case $k_g = 1$ (no guard) leads to noticeable degradation.

| $k_g$ | 1 | 2 | **3 (Default)** | 7 | 15 |
|---|---|---|---|---|---|
| FVD ↓ | 552.00 | 503.98 | **489.00** | 510.96 | 510.64 |
| FID ↓ | 39.76 | 29.62 | **26.61** | 27.55 | 29.67 |

Table 5: Static top-$p$ baselines on DrivingWorld. ENkG (default configuration from Table 3) is shown for reference. Even the best static top-$p$ setup remains substantially worse than ENkG in FVD/FID.

| Metric | $p = 0.5$ | $p = 0.7$ | $p = 0.8$ | $p = 0.9$ | $p = 1.0$ |
|---|---|---|---|---|---|
| FVD ↓ | 836.49 | 680.62 | 642.97 | 625.44 | **530.22** |
| FID ↓ | 52.64 | 46.86 | **40.03** | 47.26 | 43.73 |
| LPIPS ↓ | 0.40 | **0.36** | 0.37 | 0.38 | 0.38 |
| SSIM ↑ | 0.46 | **0.50** | 0.46 | 0.43 | 0.34 |
| PSNR ↑ | 13.50 | 15.76 | 14.75 | **16.46** | 14.66 |

### B.2 TUNING OF STATIC SAMPLING BASELINES

To further exclude the possibility that our gains come from under-tuned baselines, we perform a systematic grid search over *static* top-$p$, *static* top-$k$, and combined $pk$ sampling on DrivingWorld, while keeping all other settings (including temperature) fixed. For each baseline family, we report the *best* configuration found in the grid and compare it against ENkG.

**Static top-$p$ baselines.** We sweep over $p \in \{0.5, 0.7, 0.8, 0.9, 1.0\}$. The quantitative results are summarized in Table 5. Even under the best static configuration, the FVD remains above $530$ and FID around $40$, which are clearly worse than ENkG (FVD $= 489.00$, FID $= 26.61$).

**Static top-$k$ baselines.** We further sweep over $k \in \{30, 60, 90, 120, 150, 500\}$. Table 6 reports the results. Although FID is minimized around $k = 90$ and FVD around $k = 150$, even these best-performing static top-$k$ configurations still lag behind ENkG in both FVD and FID.

**Combined $pk$ baselines.** Finally, we explore combined $pk$ sampling (first top-$k$, then top-$p$ within the truncated set). Among the tested configurations, we find the best performance around ($p = 0.8, k = 1000$):

$$FVD = 595.93, \quad FID = 43.76, \quad LPIPS = 0.357, \quad SSIM = 0.50, \quad PSNR = 15.93.$$

While the perceptual metrics (LPIPS/SSIM/PSNR) are comparable to ENkG, the FVD and FID are still notably worse.

**Discussion.** Across all these sweeps, we always compare ENkG against the *best* static configuration of each baseline family (top-$p$, top-$k$, and $pk$). ENkG consistently achieves substantially better FVD and FID, indicating that its advantage does not come from weak or under-tuned baselines, but from the proposed *entropy-guided dynamic truncation with $k$-guard*, which provides a strictly stronger sampling strategy than any single static choice of $(p)$ or $(k)$.

Interestingly, we also observe that for very large candidate sets (e.g., $p = 1.0$ in top-$p$ or $k > 100$ in top-$k$), the generated videos can exhibit visibly fragmented or "shattered" structures, yet FVD does not necessarily increase and can even improve. This behavior is consistent with known limitations of FVD in penalizing spatial incoherence, and explains why DrivingWorld's original implementation adopts relatively defaults (e.g., $k = 30$) to preserve vehicle structural consistency rather than aggressively minimizing FVD under heavily fragmented scenes.

Table 6: Static top-$k$ baselines on DrivingWorld. The best static top-$k$ settings (e.g., $k = 90$ for FID, $k = 150$ for FVD) remain inferior to ENkG.

| Metric | $k = 30$ | $k = 60$ | $k = 90$ | $k = 120$ | $k = 150$ | $k = 500$ |
|---|---|---|---|---|---|---|
| FVD $\downarrow$ | 696.14 | 661.52 | 615.37 | 569.10 | **554.74** | 564.10 |
| FID $\downarrow$ | 61.78 | 41.54 | **34.50** | 37.86 | 39.02 | 39.10 |
| LPIPS $\downarrow$ | 0.40 | 0.39 | 0.39 | **0.37** | 0.38 | 0.38 |
| SSIM $\uparrow$ | 0.44 | 0.45 | 0.44 | **0.48** | 0.45 | 0.45 |
| PSNR $\uparrow$ | 14.04 | 14.32 | 14.05 | **15.73** | 15.34 | 15.08 |

## C QUALITATIVE RESULTS

### C.1 GENERAL-DOMAIN MODELS

To validate the effectiveness of ENkG in general domains, we evaluate it on Lumos-1 (Yuan et al., 2025) and NBP (Ren et al., 2025b). Our method mitigates error accumulation during long-horizon video generation and produces more temporally consistent results with reduced color drift and collapse artifacts. Representative qualitative results are shown in Figure 8 and Figure 9.

### C.2 ADDITIONAL COMPARISONS

We further provide more qualitative results on DrivingWorld in Figure 11 and VaVim in Figure 12.

### C.3 LONG-HORIZON GENERATION ON DRIVINGWORLD

To further evaluate ENkG under long-range autoregressive rollout, we conduct a 200-frame generation experiment on DrivingWorld. This setting is particularly prone to error accumulation and low-entropy collapse. As shown in Figure 10, ENkG substantially mitigates visual drift and maintains scene stability over very long horizons, whereas the baseline model exhibits background smearing and global color shift.

## D ANALYSIS OF ENTROPY.

### D.1 EMPIRICAL EXAMINATION OF ENTROPY IN AR VIDEO MODELS

To better understand the entropy dynamics underlying low-entropy collapse, we compare (a) the probability distributions of the top-20 tokens between a large language model (Qwen2.5 Bai et al. (2025)) and an autoregressive video model (DrivingWorld Hu et al. (2024)), and (b) the average token entropy across generation timesteps on the NuPlan dataset. As illustrated in Figure 13a, AR video models exhibit significantly sharper and more rapidly collapsing distributions, making them especially vulnerable to trajectory locking and deterministic failure modes.

### D.2 CONSEQUENCES OF THE LOW-ENTROPY TRAP

In autoregressive (AR) video models, the predictive distribution at step $t$ can be written as $p_\theta(x_t \mid x_{<t})$ with entropy $H_t = -\sum_x p_\theta(x \mid x_{<t}) \log p_\theta(x \mid x_{<t})$. We refer to a *low-entropy trap* as the regime where $H_t$ collapses prematurely and $p_\theta$ becomes pathologically overconfident around a small set of locally consistent tokens, even though the corresponding trajectory is globally suboptimal.

**Local overconfidence and trajectory locking.** Once the model enters such a regime, the effective candidate set under common sampling schemes (top-$p$, top-$k$, or $pk$) often shrinks to one or two tokens with probability mass close to 1. From that point on, the model repeatedly feeds its own highly deterministic predictions back into the context, reinforcing the same local pattern step after step. This behavior is analogous to language models falling into repetitive loops (e.g., "the the the"), persisting in a wrong chain-of-thought branch, or doubling down on an early but incorrect inference.

**Visual manifestations in video space.**   In video generation, the low-entropy trap does not merely lead to trivially frozen frames. More subtly, it manifests as: (i) **background smearing**, where large regions of the background collapse into blurry blobs and lose meaningful structural detail; (ii) **global color shift**, where the entire frame drifts toward an unnatural color cast, making consecutive frames look consistently tinted or over-/under-exposed; and (iii) **texture freezing**, where fine-grained appearance patterns (e.g., grass, water, sky) become unnaturally static and appear glued to the camera or object rather than evolving with the motion. These effects are illustrated in Figure 11 and 12, where the model settles into visually coherent yet clearly undesirable trajectories.

**Distinction from high-entropy noise.**   It is important to distinguish the low-entropy trap from the more familiar high-entropy failure mode. High entropy typically corresponds to excessive randomness, producing noisy or chaotic frames that visibly violate short-term consistency. In contrast, low-entropy collapse yields *overly deterministic* behavior: short-term consistency can even look improved, while global realism, long-horizon dynamics, and scene plausibility deteriorate.

**Effect of $k$-guard.**   Our ENkG sampler directly targets this overconfidence. When entropy falls below the lower threshold $H_{\text{low}}$, conventional truncation schemes would typically select only the single most probable token. In contrast, ENkG enforces a minimum guard size $k_g$: even in very low-entropy regimes, at least $k_g$ candidates are preserved, preventing the effective distribution from collapsing to a point mass. This mechanism maintains a controlled level of uncertainty and allows the model to explore alternative continuations that can recover from early mistakes. Empirically, we observe that enabling $k$-guard reduces the frequency and severity of texture locking, background freezing, and unnatural motion, leading to more coherent long-horizon trajectories and higher overall generative quality.

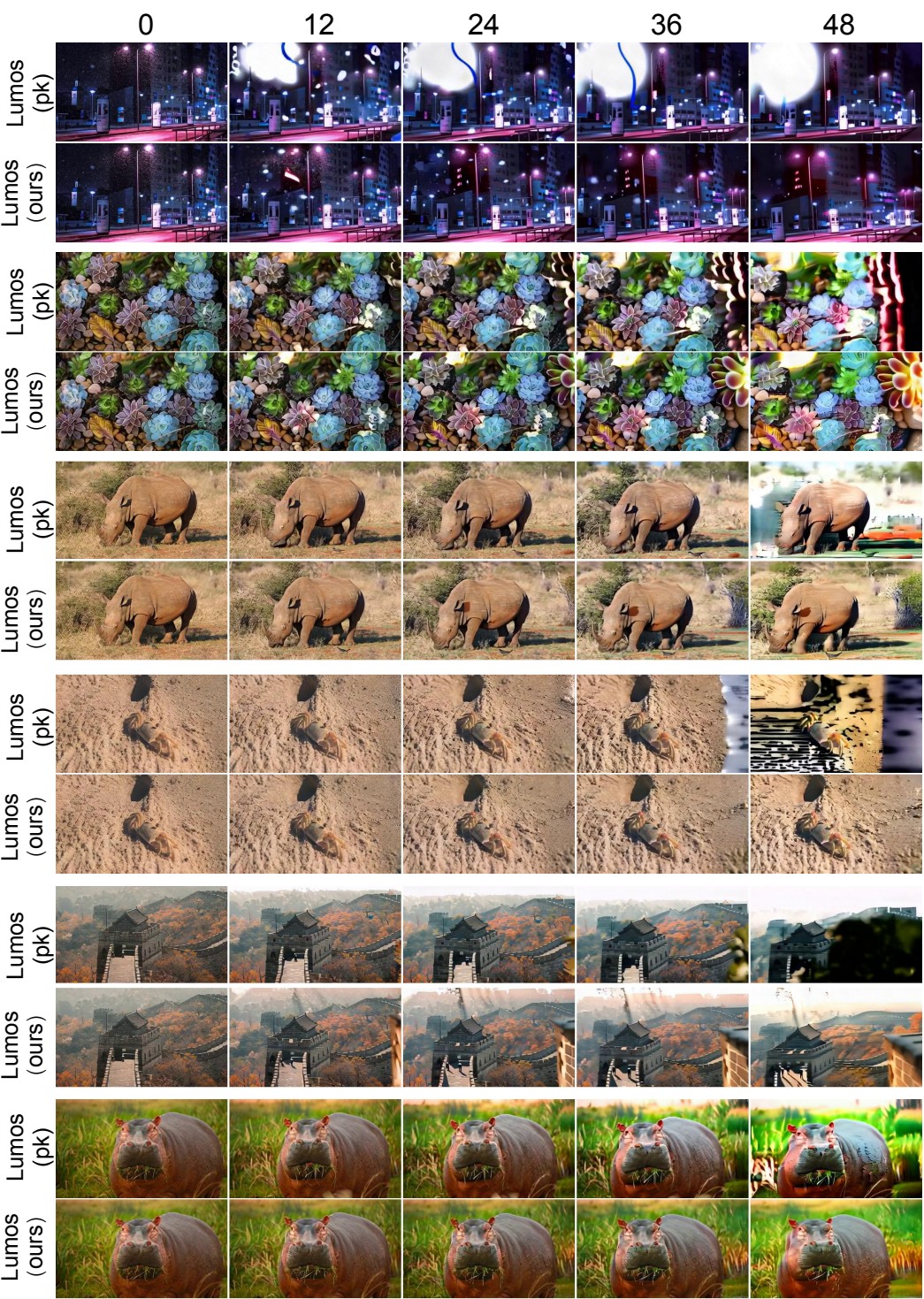

Figure 8: Comparative experiments on the *Lumos-1* model.

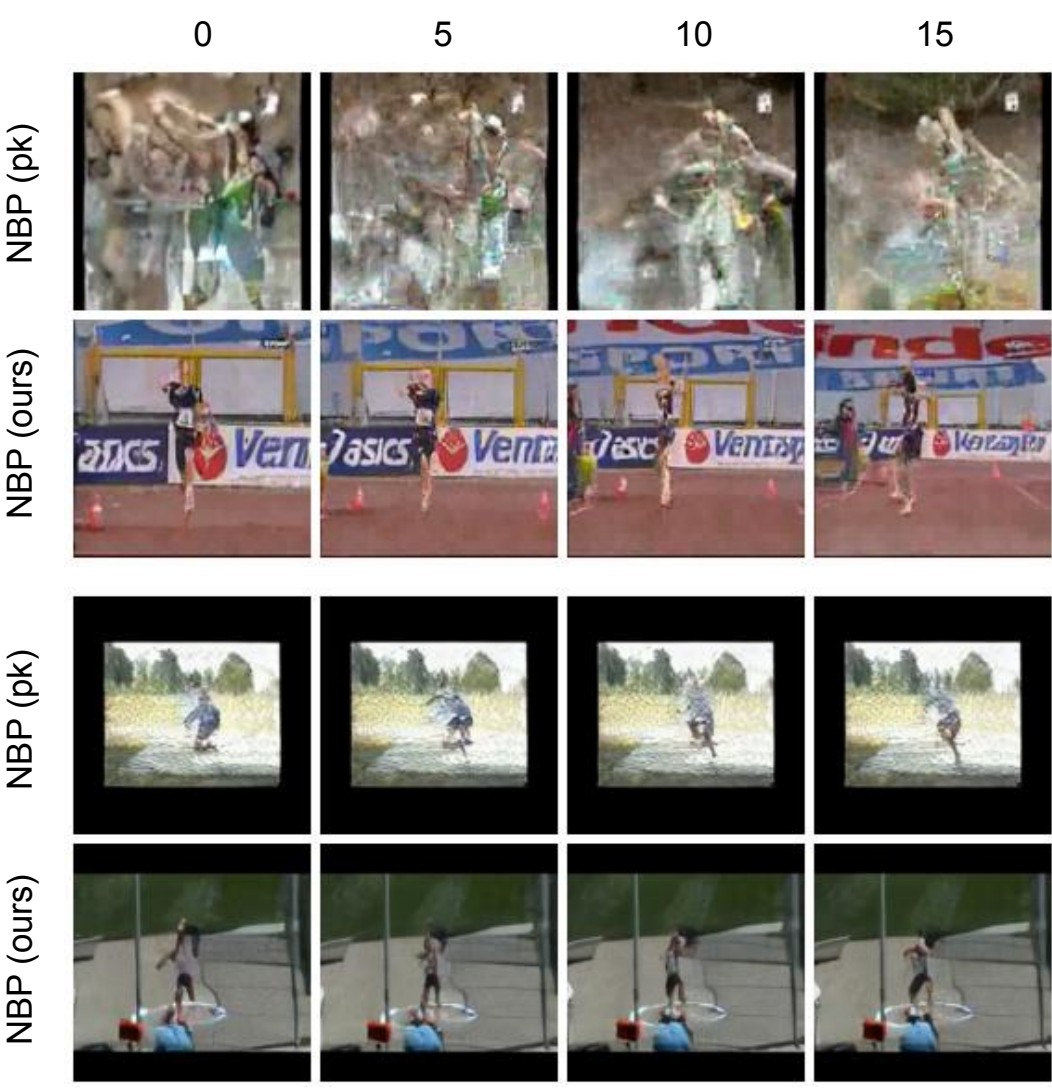

Figure 9: A Qualitative Demonstration of NBP on the UCF-101 Dataset.

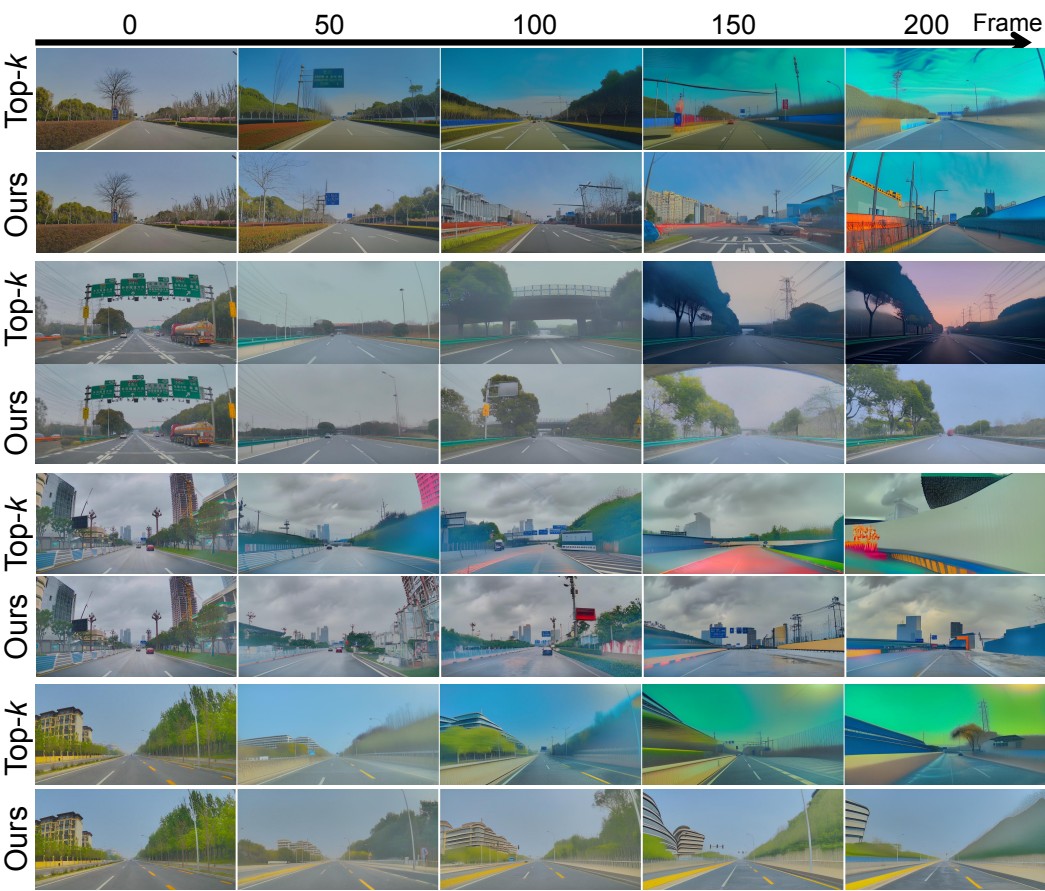

Figure 10: Long-term generation results of Drivingworld.

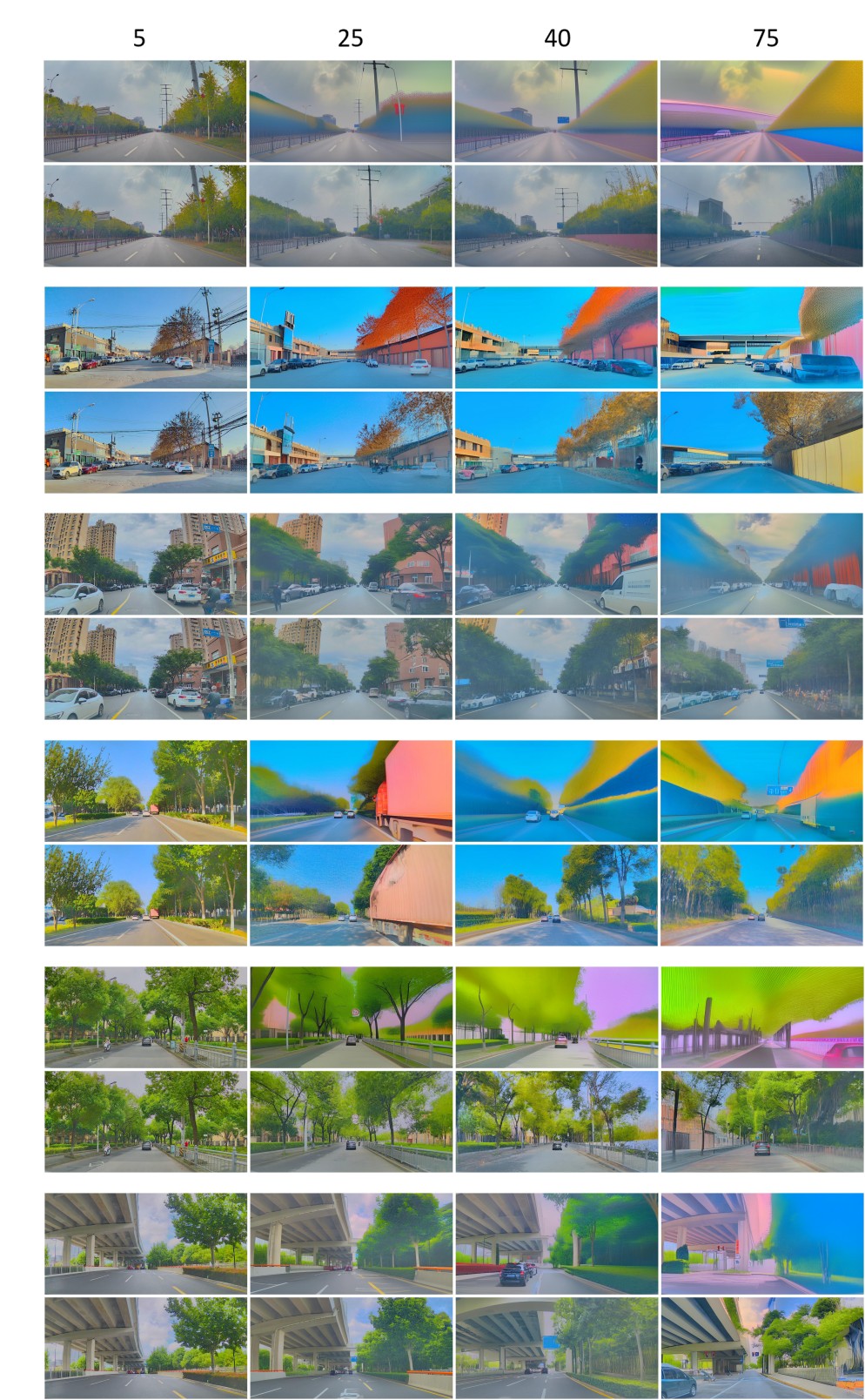

Figure 11: Additional DrivingWorld comparisons between the baseline and ENkG. (Part1)

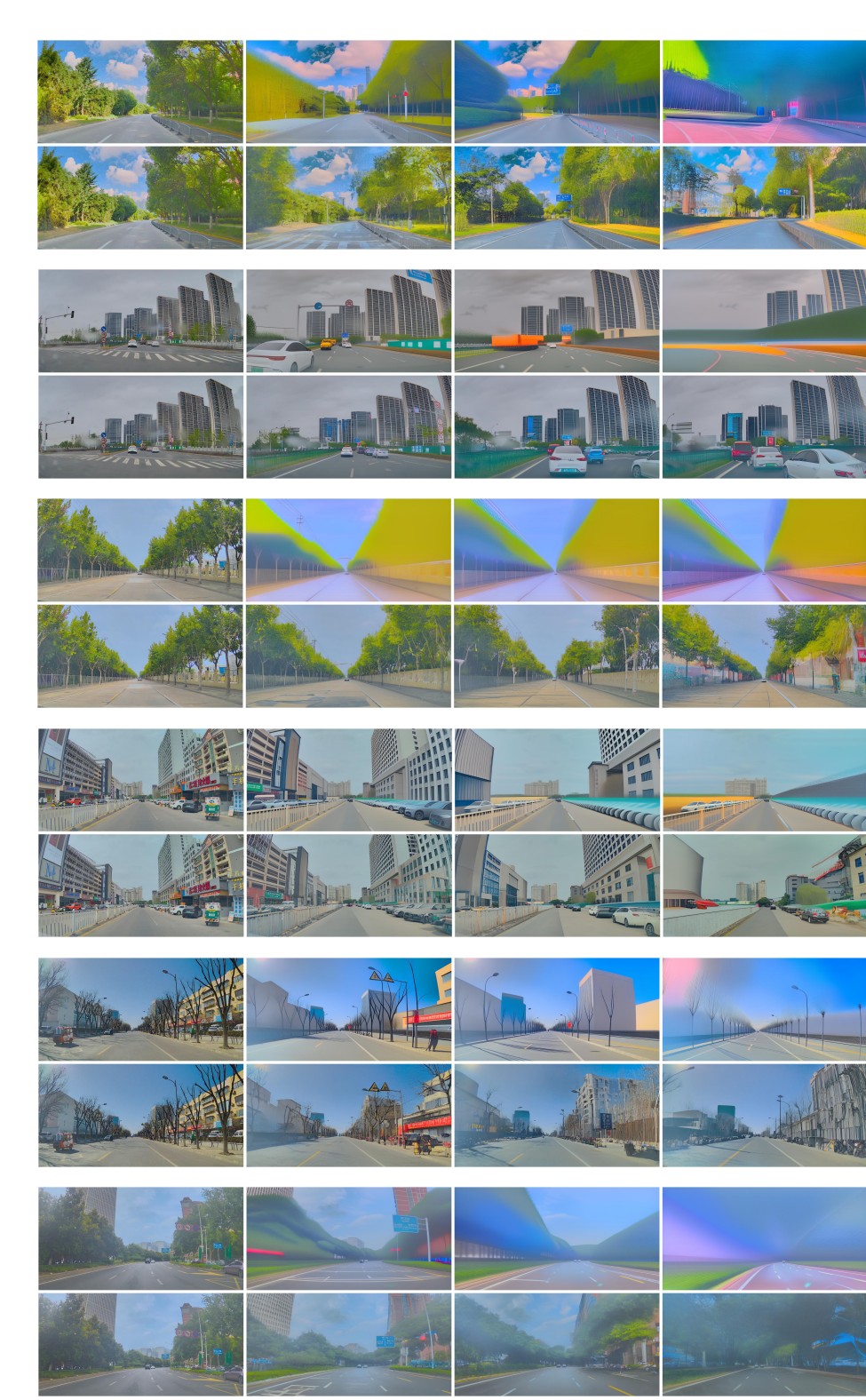

Figure 11: Part2

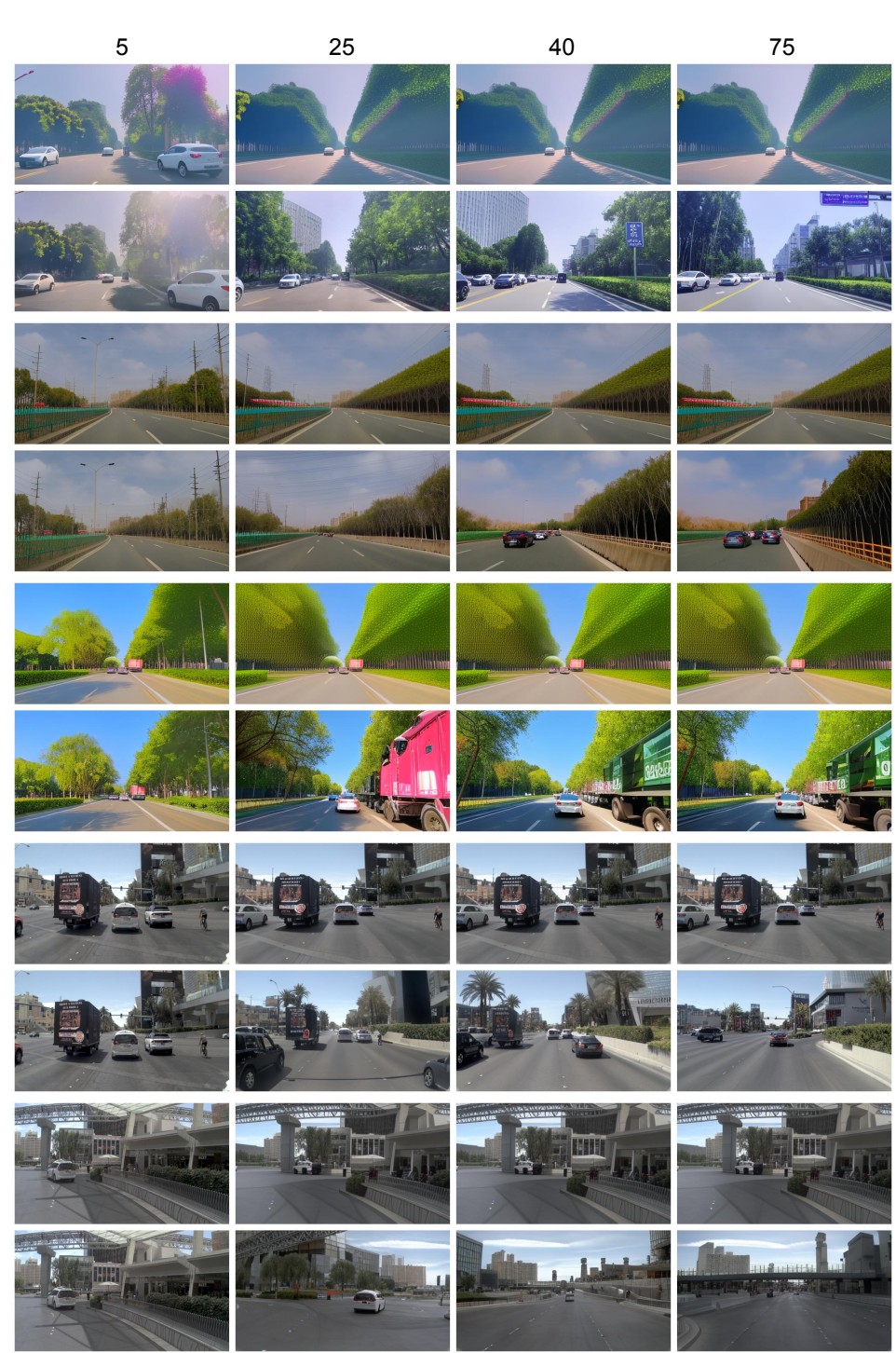

Figure 12: Additional Vavim comparisons between the baseline and ENkG.

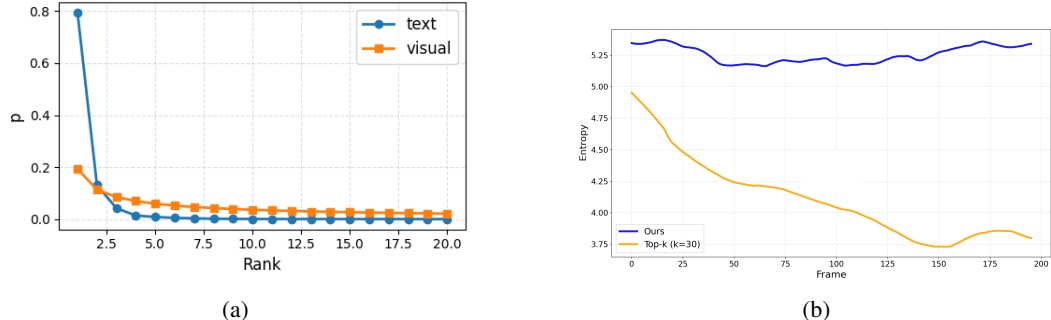

(a)                                                      (b)

Figure 13: (a).Comparison of the probabilities of the top-20 tokens between LLMs (Qwen2.5 Bai et al. (2025)) and video AR model (DrivingWorld Hu et al. (2024)). (b).Average token entropy of DrivingWorld at each frame as a function of generation timestep on the Nuplan dataset.

