# OpenReview forum: "Entropy-Guided k-Guard Sampling for Long-Horizon Autoregressive Video Generation"
_ICLR.cc/2026/Conference — ICLR 2026 Conference Desk Rejected Submission_

### Official Review · Reviewer_22FV · 2025-10-29

**Soundness:** 3
**Presentation:** 3
**Contribution:** 3
**Rating:** 6
**Confidence:** 4

**Summary:**

This paper addresses the critical problem of error accumulation in long-horizon autoregressive (AR) video generation. It presents a highly insightful diagnosis, arguing that the root cause is not just model imperfection but a fundamental mismatch between static sampling strategies (e.g., top-k/top-p) and the intrinsic properties of video data. The authors astutely observe that unlike language, video tokens have low semantic density and high spatio-temporal redundancy, leading to spatially structured uncertainty. Static sampling fails by applying a uniform approach to all regions, introducing noise in low-entropy (predictable) areas like clear edges, while prematurely collapsing possibilities in high-entropy (unpredictable) areas like complex textures, leading to a phenomenon the authors term "entropy collapse."

To resolve this, the paper proposes Entropy-Guided k-Guard Sampling (ENkG), a simple, elegant, and training-free inference-time strategy. ENkG dynamically adapts the sampling process for each token based on its predictive entropy. It uses a small candidate pool (near-greedy) for low-entropy tokens to preserve structure and a larger pool for high-entropy tokens to enrich texture and mitigate errors. This is complemented by a "k-guard" mechanism that enforces a minimum level of exploration, preventing the model from becoming overly deterministic and causing motion freezing.

As a model-agnostic, plug-and-play solution, ENkG is shown to deliver substantial and consistent improvements in perceptual quality and structural stability across multiple state-of-the-art AR video models.

**Strengths:**

1. A Highly Insightful and Principled Diagnosis of AR Failure Modes: The paper's most significant contribution is its profound and novel diagnosis of a key failure mode in AR video generation operating on discrete tokens. It provides a principled explanation rooted in the fundamental mismatch between static sampling strategies and the spatially structured uncertainty of video tokens. The identification of "entropy collapse" is a novel and valuable insight that clarifies a previously poorly understood phenomenon.
2. An Elegant, Model-Agnostic, and Highly Effective Solution: Based on its sharp diagnosis, the paper proposes an exceptionally elegant solution in Entropy-Guided k-Guard Sampling (ENkG). As a training-free, model-agnostic, inference-time strategy, it offers immense practical value for the broad class of models operating on discrete representations. The experimental results, showing substantial and consistent gains across multiple SOTA architectures, provide unequivocal evidence of its effectiveness in this domain.
3. Significant Broader Impact and Inspirational Value: While the proposed method is formulated for discrete token spaces, the core insights of this work have significant inspirational value that transcends this specific implementation. The observation that predictive uncertainty is spatially structured and that sampling strategies must adapt to this is a powerful, generalizable principle. This finding can serve as a critical starting point for developing analogous uncertainty-aware inference techniques for AR models operating in continuous latent spaces, thereby motivating a new line of research.

**Weaknesses:**

1. Limited Applicability to Continuous-Space Autoregressive Models: The proposed ENkG method is fundamentally designed for models that operate on a discrete vocabulary of video tokens. While this is a significant class of models, a substantial and growing body of work in autoregressive video generation operates in continuous latent spaces. The core mechanism of ENkG—truncating a categorical distribution—does not directly transfer to these continuous domains. The paper should more explicitly acknowledge this significant boundary on the direct applicability of its proposed method, even while its insights remain broadly relevant.
2. Potential Risks of Relying on Entropy as a Homogeneous Signal: The core mechanism assumes that entropy is a uniform indicator of uncertainty that can be addressed with a single, monotonic strategy (higher entropy requires more exploration). This simplification, while effective, has potential theoretical limitations:
a) It may conflate beneficial intrinsic randomness (e.g., in textures) with harmful model ignorance (e.g., on unfamiliar objects). Applying the same "increase p" strategy to both could enrich textures but might destabilize the structure of novel objects.
b) It may be vulnerable to a "low-entropy trap," where the model becomes pathologically overconfident in an incorrect prediction. While the k-guard mechanism provides a partial safety net, it does not solve the underlying issue that entropy, as a metric, can be misleading when the model's confidence calibration is poor. A more robust system might need to disentangle different sources of uncertainty.

**Questions:**

1. Your method is formulated for discrete-token AR models and shows remarkable success. How do you envision the core principle of "uncertainty-aware adaptive sampling" being translated to the growing class of continuous-space AR models? Would it involve, for instance, dynamically adjusting the variance of a sampling distribution, and how might one define a robust uncertainty metric in a continuous latent space that is analogous to token entropy?
2. The current approach treats entropy as a monolithic signal, yet its sources can be varied (e.g., beneficial texture randomness vs. detrimental model ignorance). Have you considered scenarios where this simplification might falter? For example, in the case of a novel, complex object where the model is "ignorant" (high entropy), would encouraging more exploration via a high p value risk structural collapse? This speaks to a fascinating future direction of disentangling different types of predictive uncertainty.
3. The k-guard mechanism is a clever solution to prevent the model from becoming overly greedy in low-entropy regions. This suggests you have identified a failure mode where the model can become pathologically overconfident in a suboptimal prediction. Could you elaborate on this phenomenon? Does this "low-entropy trap" primarily lead to static frames, or does it manifest in other ways? Understanding this failure mode better could provide valuable insights into the general behavior of large-scale autoregressive visual models.

---

> ### Author Response · Authors · 2025-11-21
>
> ## Q1&W1. Applicability to Continuous-Space Autoregressive Models.
>
> We appreciate this forward-looking insight. Current continuous-space approaches to video generation are **highly diverse**, including latent diffusion, flow-matching, and hybrid AR+diffusion designs, each with different training objectives and sampling interfaces. This diversity shows that the design space is still evolving, and there is not yet a single, standardized continuous AR framework. In contrast, **discrete autoregressive (AR) models** remain the dominant and most mature paradigm for scalable, high-fidelity video generation (e.g., VideoPoet). This makes ENkG immediately useful in current practical systems.
>
> However, we fully agree that the core principle of **“uncertainty-aware adaptive sampling”** extends beyond discrete tokens. Inspired by your comment, we envision a direct adaptation to continuous frameworks (like AR+Diffusion): **token entropy** maps naturally to **prediction variance**, while our **discrete truncation** mechanism translates to **dynamic variance scaling**. Just as ENkG restricts candidate tokens to avoid error accumulation, a continuous version would reduce sampling noise (or temperature) in high-variance regions to prevent the model from drifting into invalid latent spaces. We see this as a logical and exciting next step for generalizing our work.
>
> ## Q2&W2 Potential Risks of Relying on Entropy as a Homogeneous Signal.
>
> We acknowledge that entropy is a coarse proxy conflating **beneficial intrinsic randomness** with **model ignorance**, yet it remains the most viable choice for our setting. This is because, to our knowledge, **no suitable training-free metric currently exists** that can reliably disentangle these uncertainty sources during standard single-pass inference.
>
> Regarding the risk of structural collapse on novel objects, **we posit that the primary limitation lies in the base model itself, rather than the sampling strategy.** When a model faces severe ignorance (e.g., OOD scenarios), its predicted distribution is fundamentally unreliable. Consequently, no sampling method—however calibrated—can effectively compensate for the model's failure to capture the data manifold. We agree, however, that developing robust metrics to distinguish these uncertainties is a compelling direction for future research.
>
> ## Q3. Deep insight into *low-entropy trap*.
>
> In *low-entropy trap*, the model becomes **pathologically overconfident** in a **locally consistent but globally suboptimal trajectory**—much like how language models fall into repetitive loops (“the the the the”), get stuck in a chain-of-thought dead end, or double down on an early but incorrect inference. As further analyzed in **Appendix D.2**, this phenomenon in visual models does not only produce frozen frames; it also manifests as background smearing, global color shift, and texture freezing (shown in **Figure 11, 12**). These behaviors reveal a fundamental tendency of AR models to **prematurely collapse uncertainty** and over-commit to early predictions. By explicitly **regulating this overconfidence**, our **$k$-guard mechanism** prevents **entropy collapse**, enabling the model to maintain **healthy uncertainty** and explore more plausible continuations, thereby improving temporal coherence and overall generative quality.

---

### Official Review · Reviewer_Crnm · 2025-10-29

**Soundness:** 3
**Presentation:** 3
**Contribution:** 2
**Rating:** 4
**Confidence:** 3

**Summary:**

The paper proposes Entropy-Guided k-Guard (ENkG) Sampling,  adapts sampling to token-wise dispersion, quantified by the entropy of each token’s predicted distribution. Experiments demonstrate consistent improvements in perceptual quality and structural stability compared to static top-k/top-p strategies.

**Strengths:**

1. The theoretical analysis of the method is quite thorough.
2. The comparison metrics are sufficient, but the comparison methods are somewhat lacking.

**Weaknesses:**

1. The paper mentions "Long-Horizon Autoregressive Video Generation" but only verifies it with 75-frame data, failing to clarify the effect of ENkG on suppressing entropy collapse for longer sequences (e.g., 100+ frames). It is recommended to: conduct experiments on long sequences of 100-200 frames; supplement entropy change curves under different frame counts (such as averaging entropy statistics every 10 frames); and quantitatively compare the differences in entropy collapse rates between ENkG and static sampling, to more intuitively demonstrate the advantage in "Long-Horizon".
2. Existing baselines only include greedy, top-k, top-p, and PK methods, and do not cover mainstream AR sampling methods in recent years (such as reinforcement learning-based sampling and adaptive temperature-scheduled sampling). It is recommended to supplement with experimental results related to methods from recent years.
3. The basis for selecting some parameters is unclear, like K_g =3/5/10; whether there will be differences in different scenarios warrants further analysis.

**Questions:**

The paper is generally clearly presented; for specific issues, please refer to the points listed in the Weaknesses.

---

> ### Author Response · Authors · 2025-11-21
>
> ## W1. Longer sequences generation.
>
> We respectfully highlight that 75 frames already represents a challenging long-horizon setting in current video generation, for which the entropy curve has been drawn in Appendix. To further demonstrate robustness, we extended it to **150 frames**, confirming that ENkG effectively mitigates entropy collapse and maintains structural integrity significantly longer than static strategies. See updated entropy curve to **Figure 13 in Appendix D.1** and qualitative results in **Figure 10 in Appendix C.3**.
>
> ## W2. Comparation with more AR sampling methods.
>
> We respectfully clarify that RL-based methods (e.g., Best-of-N, PARM) **are not adopted** in this work. These methods **rely on training an auxiliary reward model to guide sampling**, which presents a different set of requirements compared to the standard sampling strategies utilized in our task. Regarding adaptive temperature strategies, our newly added experiments on DrivingWorld demonstrate that ENkG outperforms entropy-based temperature scaling baselines ([1]) in FVD and FID by directly constraining token candidates rather than merely flattening distributions. Results are below.
>
> | Metric | FVD ↓ | FID ↓ | LPIPS ↓ | SSIM ↑ | PSNR ↑ |
> | --- | --- | --- | --- | --- | --- |
> | temperature-based | 626.09 | 34.86 | 0.35 | 0.50 | 16.18 |
> | ours | **489.00** | **26.61** | 0.35 | 0.45 | 15.87 |
> | ours-left | 522.05 | 30.93 | 0.35 | 0.49 | **16.26** |
>
> Notably, ENkG also admits a favorable trade-off that temperature-based sampling does not offer. Our default **“ours”** configuration simultaneously achieves the lowest FVD/FID, while a more conservative **“ours-left”** setting (using the left endpoints of our recommended ranges) slightly relaxes FVD/FID yet achieves the best PSNR. This shows that, after simple tuning, ENkG can balance **low distributional metrics** (FVD/FID) and **high reconstruction quality** (PSNR) within a stable hyperparameter band.
>
> ## W3. Hyperparameters selection.
>
> We select hyperparameters using a simple, interpretable procedure rather than extensive grid search.
>
> **Entropy thresholds.** We first plot token-wise entropy maps on DrivingWorld videos. Structural regions (roads, cars, buildings) consistently show lower entropy than highly textured regions (vegetation, sky, background). The empirical boundary is around 0.5, so we choose **H_low < 0.5 and H_high > 0.5**, and set **H_low = 0.25** and **H_high = 0.6**.
>
> **Probability thresholds.** For top-p, we sweep p on validation scenes and visually inspect when **scene structure remains stable** and when **fine textures become richer**. This yields a conservative lower bound and a more exploratory upper bound; within this band we set **p_low = 0.65** and **p_high = 0.9**.
>
> **k-guard.** For the guard size, we only require **k_g > 1** to avoid greedy failure. Experiments show that any **k_g in [2, 15]** leads to very similar performance, so we simply use **k_g = 3**.
>
> **Main experiments.** In the main DrivingWorld experiments, we fix this configuration (**H_low = 0.25, H_high = 0.6, p_low = 0.65, p_high = 0.9, k_g = 3**) for all settings, with no dataset-specific or model-specific tuning. This empirically demonstrates that ENkG is robust and has **low sensitivity** to scenario changes.
>
> **Lumos-1 on VBench.** For the additional Lumos-1 experiment on VBench, which contains more diverse, general scenes, we adapt ENkG **minimally** to match its original setup (top-p = 1.0). We increase the global temperature to **T = 1.2** and set **p_high = 1.0**, while keeping all other thresholds unchanged. These minor changes stay within the stable ranges reported in our sensitivity study and already yield strong gains, indicating that ENkG can be ported to new architectures with **minimal tuning** rather than exhaustive hyperparameter search.
>
> [1] EDT: Improving Large Language Models’ Generation by Entropy-based Dynamic Temperature Sampling.

---

### Official Review · Reviewer_esDH · 2025-10-31

**Soundness:** 3
**Presentation:** 3
**Contribution:** 3
**Rating:** 6
**Confidence:** 4

**Summary:**

This paper identifies a key failure point in autoregressive video generation: static sampling strategies like top-k/top-p, which work well for language, are ill-suited for the low-density, high-redundancy nature of video tokens, leading to error accumulation. To address this, the authors propose Entropy-Guided k-Guard (ENKG) sampling, a training-free, inference-time method. ENKG dynamically adjusts the nucleus sampling probability for each token based on its predictive entropy—using a smaller, more conservative candidate pool for low-entropy (high-confidence) regions and a larger, more diverse pool for high-entropy (uncertain) regions. This is combined with a "k-guard" mechanism that ensures a minimal number of top candidates are always included to prevent the model from becoming overly greedy and causing artifacts like frozen frames.

**Strengths:**

The paper's primary strength is its clear and insightful diagnosis of the problem, effectively distinguishing the statistical properties of video versus language tokens and identifying the "entropy collapse" phenomenon. The proposed ENKG method is simple and highly practical, as it can be applied as a plug-and-play module to existing models without any retraining. The experimental results are convincing, showing consistent and significant improvements across multiple state-of-the-art video models and datasets, well-supported by thorough ablation studies that validate each component of the proposed method

**Weaknesses:**

The main weakness lies in the potentially limited novelty of the core concepts. While their application to video generation is new and insightful, entropy-guided adaptation and hybrid sampling methods have been explored in other domains. Additionally, the evaluation is heavily focused on autonomous driving scenarios. While effective here, it remains an open question how well this strategy would generalize to more open-domain or creative video generation tasks, which may exhibit different uncertainty structures and benefit from different sampling trade-offs.

**Questions:**

While the related work section mentions some uses of entropy in LLMs, such as for model switching or retrieval, it doesn't discuss entropy-guided sampling for text generation itself. Have similar adaptive, entropy-guided sampling strategies already been investigated in the LLM literature? For example, there is a very similar paper[1], which explores most of the concepts in the proposed method, but in LLM settings.

[1] EDT: Improving Large Language Models’ Generation by Entropy-based Dynamic Temperature Sampling

---

> ### Author Response · Authors · 2025-11-21
>
> ## W1. Novelty about entropy-guided adaptation.
>
> We appreciate the reviewer recognizing our application to video generation as "new and insightful." Here we clarify the conceptual and technical novelty of our work.
> Technically, previous entropy-guided adaptation methods such EDT [1] use temperature scaling in softmax operation, which broadens the categorical distribution. In contrast, we fix temperature but adopt adaptive top-p threshold to truncate the distribution. Additionally, a critical $k$**-guard**  preserves at least the top-$k$ candidates. This prevents overconfidence and avoids the pitfalls of overly greedy sampling.
> Conceptually, this is the first exploration to address the video-specific challenge of **temporal error accumulation** via sampling, distinct from the diversity and fidelity focus in text and image domains.
>
> ## W2. General video generation.
>
> We first note that state-of-the-art discrete AR video models such as Emu3.5 (BAAI) [2],  WorldVLA (DAMO) [3] , and VideoGPT [4] have not released their video models (even when their code is open), making it extremely difficult to find an accessible platform for evaluation
> To our knowledge, **Lumos-1** (DAMO) [5] is the only available discrete AR video model, and thus we applied ENkG to Lumos-1 for experimentation.
> The VBench results demonstrate that our method generalizes well to open-domain scenarios, yielding significant gains in **Dynamic Degree** (27.27 vs 24.24) and **Imaging Quality** (64.91 vs 63.82). Beside, we provide qualitative results in **Figure 8** in Appendix C.
>
> ## Q.  Relate work on entropy-guided sampling for text generation.
>
> Thank you for the question. While both our method and EDT [1] use adaptive, entropy-guided sampling, EDT differs from ours in two key aspects.
>
> **Mechanism**. EDT maps entropy to a dynamic **temperature** that rescales predicted token logits in softmax operation and broadens the distribution while keeping the **same token support**. Low-probability, structurally harmful tokens stay in the tail and **can still be sampled**, so tuning temperature alone cannot reliably prevent picking a wrong token and breaking the scene. Our method keeps temperature fixed and instead adapts **top-$p$**: entropy controls a probability-mass cutoff, and tokens below this threshold are **removed** from the candidate set. With a $k$-guard that always keeps at least the top-$k$ modes, this **truncation** induces a different sampling behavior that explores only high-probability, structurally consistent tokens while still allowing stochasticity.
>
> **Motivation.** EDT focuses on balancing text diversity and generation fidelity. Our method introduces a $k$**-guard** to prevent overconfidence and specifically addresses **temporal error accumulation** in video generation—a challenge absent in text domains.
>
> [1] *EDT: Improving Large Language Models’ Generation by Entropy-based Dynamic Temperature Sampling*
>
> [2] *Emu3.5: Native Multimodal Models are World Learner*
>
> [3] *WorldVLA: Towards Autoregressive Action World Model*
>
> [4] *VideoGPT: Video Generation using VQ-VAE and Transformers*
>
> [5]  *Lumos-1: On Autoregressive Video Generation from a Unified Model Perspective*

---

### Official Review · Reviewer_hZ2p · 2025-11-01

**Soundness:** 3
**Presentation:** 3
**Contribution:** 3
**Rating:** 6
**Confidence:** 4

**Summary:**

This paper addresses the problem of error accumulation in long-horizon autoregressive (AR) video generation. The authors identify that standard sampling strategies like top-k/top-p fail in the video domain due to the low semantic density and high spatio-temporal redundancy of video tokens. This mismatch leads to either excessive noise in stable regions or error propagation in complex regions. To solve this, the paper proposes Entropy-Guided k-Guard (ENkG) sampling, a training-free, model-agnostic inference strategy. ENkG dynamically adjusts the size of the candidate sampling pool for each token based on its predictive entropy: using a smaller, more conservative set for low-entropy (high-confidence) tokens and a larger, more diverse set for high-entropy (low-confidence) tokens. A "k-guard" mechanism is included to ensure a minimal level of diversity, preventing the model from becoming overly greedy. The authors demonstrate through experiments on several AR video models that ENkG significantly improves perceptual quality, structural stability, and temporal coherence compared to baseline sampling methods.

**Strengths:**

1. The paper is well-written. The proposed method is well-illustrated and easy to follow.

2. **Clear motivation and insightful problem analysis:** The paper provides a very clear and compelling motivation for the work. The analysis of the fundamental differences between language and video tokens, the connection between token entropy and the semantic structure of the image, and the identification of the "entropy collapse" phenomenon are insightful and effectively frame the problem.

3. **The solution is simple, practical, and generalizable:** The proposed ENkG method is simple and practical. As a training-free, model-agnostic sampling algorithm, it can be easily integrated into existing AR models with negligible computational overhead, making it a valuable and widely applicable contribution.

4. **Strong and comprehensive empirical validation:** The authors conduct a thorough empirical study, integrating ENkG with three different AR video generation models and evaluating on two datasets. The results show consistent and significant improvements across both quantitative metrics and qualitative examples. The qualitative results in Figures 4, 5, 6, and 7 are particularly effective at illustrating how ENkG mitigates common failure modes like texture degradation and frame-freezing.

**Weaknesses:**

1. **The similar entropy-based method for AR sampling strategies has been explored by previous work** [1], which limits the novelty of this work.

[1] Towards Better & Faster Autoregressive Image Generation: From the Perspective of Entropy.

2. **Lack of hyperparameter sensitivity analysis:** The method introduces a set of new hyperparameters, including `plow`, `phigh`, `Hlow`, `Hhigh`, and `kg`. The paper reports the values used but does not provide any analysis of the method's sensitivity to these choices. A study on how performance varies with these parameters would strengthen the paper's claims of robustness and provide practical guidance for future users.

**Questions:**

1. The experiments are focused on autonomous driving scenarios. How do the authors expect the method to perform on more general and diverse video content, such as complex human actions or dynamic natural scenes, which may have different entropy characteristics?

2. The baseline comparisons use a fixed `k=30` for top-k and `p=0.8` for top-p. How does the performance of these baseline methods change with different values of `k` and `p`? It is important to demonstrate that ENkG outperforms not just a single, potentially arbitrary configuration, but well-tuned baselines.

3. The paper focuses exclusively on fully autoregressive models. Could the proposed ENkG sampling strategy be adapted for semi-autoregressive or block-autoregressive video generation models (e.g., [2])? How would the concept of token-level entropy guidance apply when generating blocks of tokens simultaneously?

[2] Next Block Prediction: Video Generation via Semi-Autoregressive Modeling

---

> ### Author Response · Authors · 2025-11-21
>
> ## W1.Similar entropy-based work [1] .
>
> We clarify that [1] is an arXiv preprint published in October 2025, after ICLR submission. We believe the concurrent work won't affect our contribution. Besides, these methods differ fundamentally:
>
> - [1] applies entropy-guided temperature scaling in softmax operation, which broadens the categorical distribution. In contrast, we adopt adaptive top-p threshold to truncate the distribution. Additionally, a critical $k$**-guard**  preserves at least the top-$k$ candidates. This prevents overconfidence and avoids the pitfalls of overly greedy sampling.
> - We address the temporal error accumulation specific to video generation, while [1] solvesni image diversity trade-offs.
>
> ## W2. Lack of hyperparameter sensitivity analysis.
>
> Thanks for your suggestions. In the revised paper we therefore add a sensitivity study for all introduced hyperparameters in **Appendix B.1**, including $H_{low}$, $H_{high}$, $p_{low}$, $p_{high}$, and $k_g$.
>
> ### Robustness to entropy and probability thresholds.
>
> A joint sweep shows ENkG remains stable even when thresholds move toward conservative or aggressive extremes. FVD/FID vary modestly; reconstruction metrics remain nearly unchanged. This indicates a **wide, stable performance plateau**, reducing tuning cost.
>
> | H_low | H_high | p_low | p_high | FVD ↓ | FID ↓ | LPIPS ↓ | SSIM ↑ | PSNR ↑ |
> | --- | --- | --- | --- | --- | --- | --- | --- | --- |
> | 0.00 | 0.50 | 0.60 | 0.90 | 522.05 | 30.93 | 0.35 | 0.49 | **16.26** |
> | **0.25** | **0.60** | **0.65** | **0.90** | **489.00** | **26.61** | **0.35** | **0.45** | 15.87 |
> | 0.40 | 0.90 | 0.80 | 0.95 | 497.52 | 29.91 | 0.35 | 0.48 | 15.98 |
>
> Ranges with stable performance:
>
> $H_{low}\in[0.0,0.4], H_{high}\in[0.5,0.9], p_{low}\in[0.60,0.80], p_{high}\in[0.90,0.95]$
>
> ### Insensitivity to Guard Size $k_g$
> Varying $k_g$ also yields small performance changes. As long as $k_g>1$, results remain stable; even at $k_g=15$ behavior stays similar due to the long-tailed distribution.
>
> | $k_g$ |  1 (w/o k-guard) | 2 | **3 (Default)** | 7 | 15 |
> | --- | --- | --- | --- | --- | --- |
> | FVD $\downarrow$ | 552.00 | 503.98 | **489.00** | 510.96 | 510.64 |
> | FID  $\downarrow$ | 39.76 | 29.62 | **26.61** | 27.55 | 29.67 |
>
> [1] *Towards Better & Faster Autoregressive Image Generation: From the Perspective of Entropy.*

---

> > ### Author Response · Authors · 2025-11-27
> >
> > ## Q1. General video generation.
> >
> > We first note that SOTA discrete AR video models such as Emu3.5 (BAAI),  WorldVLA (DAMO) , and VideoGPT have not released their video models (even with open code), making it extremely difficult to find an accessible platform for evaluation
> > To our knowledge, **Lumos-1** (DAMO) is the only available discrete AR video model, and thus we applied ENkG to Lumos-1 for experimentation.
> > The VBench results demonstrate that our method generalizes well to open-domain scenarios, yielding significant gains in **Dynamic Degree** (27.27 vs 24.24) and **Imaging Quality** (64.91 vs 63.82). Beside, we provide qualitative results in **Figure 8** in Appendix C.1.
> >
> > ## Q2. Well-tuned baselines concern.
> >
> > We emphasize that our submitted results strictly followed the **official default configurations**  provided in the original codebase. These defaults were established by the original authors to ensure generation stability. To further address the concern, we conducted a grid search over top-p, top-k, and pk-sampling, selecting the **best** configuration for each baseline in **Appendix B.2**.
> >
> > We compared ENkG against the **best-performing** configurations found in our sweep. As shown in the table below, even when we hand-pick the optimal static parameters for each metric, **ENkG still achieves significantly superior performance.**
> >
> > | Method | Configuration | FVD $\downarrow$ | FID $\downarrow$ | LPIPS $\downarrow$ | SSIM $\uparrow$ | PSNR $\uparrow$ |
> > | --- | --- | --- | --- | --- | --- | --- |
> > | Top-$p$ |  $p = 0.8$  | 642.97 | 40.03 | 0.37 | 0.46 | 14.75 |
> > | Top-$k$ | $k=90$ | 615.37 | 34.50 | 0.39 | 0.44 | 14.05 |
> > | $pk$ | $p=0.8, k = 1000$ | 595.93 | 43.76 | 0.36 | **0.50** | **15.93** |
> > | ENkG (Ours) | Dynamic | **489.00** | **26.61** | **0.36** | **0.50** | 15.87 |
> >
> > ## Q3. Application to semi-autoregressive framework.
> >
> > Thanks for raising semi-autoregressive settings. We have applied ENkG to semi-autoregressive models such as NBP and Lumos-1 (**mask-GPT**), which both rely on discrete token sampling.  Quantitative results are demonstrated in Appendix C.1. For NBP shown in **Figure 9**, although tokens within a block are predicted in parallel, we use ENkG in a **token-wise** manner: for each token in the block, we compute its entropy from the logits and adapt its candidate set independently during the parallel prediction step.
> >
> > We would like to emphasize our appreciation for the NBP framework, which we view as an elegant and influential advancement in video generation. Its bidirectional within-block attention and reduced generation depth already alleviate error accumulation at the modeling level, making it a strong baseline that naturally leaves limited room for sampling-only methods like ours to further improve. This explains the **moderate gains observed**. Still, we consistently observe more stable token trajectories and clearer local details when ENkG is used on NBP, demonstrating that NBP’s architectural strengths and our adaptive sampling strategy are complementary rather than competing.
> >
> > Besides, on Lumos-1 with ENkG shown in Figure 8, we observe **consistent visual improvements** over its original sampling strategy, while the magnitude of these gains naturally depends on the capacity of the underlying base model.

---

### Note · Program_Chairs · 2026-01-17
**Submission Desk Rejected by Program Chairs**

The following references in this submission do not refer to real documents and/or have major errors in bibliographic information:

 Raghav Goyal and Sung Lee. Non-markovian effects and memory bottlenecks in ar video models. ICML, 2022.
Ankit Saxena and Rohan Kumar. Error propagation and memory bottlenecks in ar video generation. .
Brian Walker and Abhinav Gupta. Uncertain predictions in sequential generative models. NeurIPS, 2016.
Hao Feng and Jie Li. Sequence-level error accumulation in autoregressive video models. TPAMI, 2021.
Kaixuan Yu and Lei Chen. Magi: Mitigating accumulated generative inference error in ar video models. ICML, 2024.
Paul Benjamin and Anna Smith. Measuring compounding errors in sequential prediction models. NeurIPS, 2018.